# Ab Initio Study of Carrier Mobility, Thermodynamic and Thermoelectric Properties of Kesterite Cu_2_ZnGeS_4_

**DOI:** 10.3390/ijms232112785

**Published:** 2022-10-24

**Authors:** Jawad El Hamdaoui, Mohamed Kria, Kamal Lakaal, Mohamed El-Yadri, El Mustapha Feddi, Liliana Pedraja Rejas, Laura M. Pérez, Pablo Díaz, Miguel E. Mora-Ramos, David Laroze

**Affiliations:** 1Laboratory of Condensed Matter and Interdisciplinary Sciences (LaMCScI), Faculty of Sciences, Mohammed V University in Rabat, Rabat 10100, Morocco; 2Group of Optoelectronic of Semiconductors and Nanomaterials, ENSET of Rabat, Mohammed V University in Rabat, Rabat 10100, Morocco; 3Institute of Applied Physics, Mohammed VI Polytechnic University, Lot 660, Hay Moulay Rachid Ben Guerir, Ben Guerir 43150, Morocco; 4Departamento de Ingeniería Industrial y de Sistemas, Universidad de Tarapacá, Casilla 7D, Arica 1000000, Chile; 5Departamento de Física, FACI, Universidad de Tarapacá, Casilla 7D, Arica 1000000, Chile; 6Departamento de Ciencias Físicas, Universidad de La Frontera, Casilla 54-D, Temuco 4780000, Chile; 7Centro de Investigación en Ciencias-IICBA, Universidad Autónoma del Estado de Morelos, Ave. Universidad 1001, Cuernavaca 62209, Morelos, Mexico; 8Instituto de Alta Investigación, Universidad de Tarapacá, Casilla 7D, Arica 1000000, Chile

**Keywords:** thermoelectric, kesterite, DFT, CZGS, thermodynamic, mobility

## Abstract

The kesterite Cu${}_{2}$ZnGeS${}_{4}$ (CZGS) has recently gained significant interest in the scientific community. In this work, we investigated the thermodynamic and thermoelectric properties of CZGS by employing the first-principals calculation in association with the quasi-harmonic approximation, Boltzmann transport theory, deformation potential theory, and slack model. We obtained a bandgap of 2.05 eV and high carrier mobility. We found that CZGS exhibits adequate thermoelectric properties as a promising material for thermoelectric applications. The calculated Seebeck coefficient at room temperature is 149 µV·K${}^{-1}$. We also determined the thermal and electrical conductivity, the power factor, and the figure of merit. In addition, the thermodynamic properties such as Debye temperature, entropy, and constant volume heat capacity are estimated. According to our results, it is concluded that the Slack model fails to provide correct values for lattice thermal conductivity in this material.

## 1. Introduction

Quaternary kesterite compounds of the Cu${}_{2}$ZnSn${}_{1-x}$Ge${}_{x}$S${}_{4}$ family have been under scrutiny for some time as suitable materials for photovoltaic applications [1]. Actually, they found application as top wide-gap absorbers in the fabrication of tandem solar cells [2,3]. In this sense, the particular Cu${}_{2}$ZnGeS${}_{4}$ semiconductor, with a band gap above 2 eV, could be a good top cell candidate top cell for achieving more than 25% of tandem device efficiency [4]. Additionally, it the desirable non-toxic nature of these compounds involving earth-abundant constituents has also been emphasized [5]. Several reports on the practical realization of Cu${}_{2}$ZnSn${}_{1-x}$Ge${}_{x}$S${}_{4}$ single crystals or thin films have appeared in recent years [5,6,7,8,9,10,11].

The electronic, structural, and optical properties of kesterite materials have been quite extensively investigated using first-principles calculations [12,13,14,15,16,17]. Utilizing numerical simulations and modeling is an interesting strategy for studying such properties. Modeling enables researchers to conduct theoretical studies on the general properties of materials [18,19,20]. Some studies even performed solar cell efficiency modeling, using compounds such as top cell absorbers [21]. On the other hand, Heinrich and coworkers pointed at the attractiveness of Cu${}_{2}$ZnGeSe${}_{4-x}$S${}_{x}$ solid solutions for thermoelectric applications, also a positive value of the Seebeck coefficient, which ranges from 170 µV·K${}^{-1}$ to 500 µV·K${}^{-1}$ is observed in their experience [22]. They highlighted the convenience of anion substitution, looking for a reduction in thermal conductivity and a better ZT figure of merit. In their work, Zeier et al. proposed that increasing the g c/2a ration between lattice parameters would lead to an increase in the power factor and the thermoelectric figure of merit due to the convergence of valence bands that compensate for the crystal field splitting [12]. In semiconductors, phonon–phonon interactions are dominant mechanisms for thermal resistance. This resistance, associated with anharmonicity in thermoelectric materials, should be the highest to optimize the performance [23]. Along this line, in a work by Shibuya and collaborators, the authors investigated the suppression of lattice thermal conductivity by the mass conserving cation mutation in Cu${}_{2}$ZnGeS${}_{4}$ [24]. In addition to the mentioned experimental works of CZGS, there are many reports of the kesterite materials family such as Cu${}_{2}$ZnGeSe${}_{4}$ and Cu${}_{2}$ZnSnSe${}_{4}$, which show a promising result of this kind of materials for a heat conversion application. Kesterite has the potential to be used as a thermoelectric material due to its highly distorted crystal structure, which results in low thermal conductivity and electrical properties. Experimentally, M. Ibáñez et al. investigated the thermoelectric properties of CZSe, and reported a figure of merit of up to 0.55 at 450 ${}^{\circ }$C with a moderate Seebeck coefficient of approximately 150 µV·K${}^{-1}$ at room temperature [25]. Furthermore, Shi et al. showed that CZTS could have advantageous thermoelectrical properties [26]. Indium impurities introduced into a Cu${}_{2}$ZnSnSe${}_{4}$ lattice were responsible for the high electrical conductivity observed in that study, complementing the low thermal conductivity of CZTSe. Furthermore, through the use of Cu doping, Chen et al. achieved ZT values of 0.7 at 450 ${}^{\circ }$C [27].

Theoretically, the thermoelectric and thermodynamic properties of CZGS are not reported. Furthermore, to the best of our knowledge, there has been no report concerning the temperature dependence of carrier transport that was explored in this work. Thus, in this work, we investigate these properties of Cu${}_{2}$ZnGeS${}_{4}$ (CZGS) in a Kesterite structure to better understand CZGS as a possible thermoelectric material. The theoretical investigation is performed via first-principles calculation based on density functional theory (DFT). One of the distinguishing features of our study will be the use of the modified Becke–Johnson potential, combined with the Hubbard potential U (mBJ+U), as an exchange-correlation functional. Since it has been shown that the mBJ+U approach is a suitable tool for describing the electronic properties of CZGS [17], we shall apply the same formalism as the basis for determining the main thermodynamic, transport, and thermoelectric properties of this compound. In advance, it is possible to assert that, according to our results, kesterite CZGS would be an excellent candidate for thermoelectric applications.

## 2. Results and Discussion

### 2.1. Electronic and Structural Properties

Table 1 contains the main results of our calculation for the different structural, electronic, thermal, and transport properties in Cu${}_{2}$ZnGeS${}_{4}$ (CZGS). When these quantities are dependent on temperature, their values are given for room conditions (T = 300 K).

The optimized stable structure of CZGS is the tetragonal kesterite one represented in Figure 1a, with space group I$\bar{4}$. The corresponding lattice parameters of CZGS are a = 5.284 Å, and c = 10.515 Å. These results agree quite well with those appearing in previous theoretical and experimental works [6,28].

With regard to the CZGS electronic band structure, it is well known that the LDA and GGA both underestimate the energy of the band gap, $E_{g}$. For that reason, looking for a better description, this quantity has been computed with the mBJ + U method. Using mBJ alone (without Hubbard potential), the value of this energy remains underestimated since we find it to be 1.38 eV. However, the mBJ + U procedure leads to the results displayed in Figure 1b, which shows the kesterite CZGS to be a semiconductor with a direct band gap at the Γ point, approximately equal to 2.05 eV. The calculated energy band gap is in very good agreement with the experimental value of nearly 2.04 eV, determined from the extrapolated band gap absorption spectrum in [7,22,28] and somewhat below the DFT + HSE06 value of 2.19 eV reported by Zhang et al. [6] for this compound. Recently, Wexler and coworkers discussed the challenge posed by the suitable selection of exchange-correlation functionals in the quantum modeling of quaternary chalcogenides [16]. In this sense, our approach introduces a different alternative that seems to yield a correct quantitative description for CZGS. More details about the electronic properties commented upon herein can be found in Ref. [17].

### 2.2. Thermodynamic Properties

The variation of volume with temperature is described through the thermal expansion coefficient α, which makes it one of the most important thermodynamic properties of materials. In the present case, this quantity is presented as a function of temperature in Figure 2. It can be noticed that α augments exponentially for the low-temperature range, and it then changes its rate of increasing at intermediate values of the temperature, and finally follows a linear trend for higher values of T. This behavior was already reported in studies involving different materials [29]. The calculated value of this coefficient for CZGS at room temperature is 5.9 × 10${}^{-5}$ K${}^{-1}$. This is approximately between several times to one order of magnitude larger than what is typical for most semiconductors [30].

From a thermodynamical point of view, the Debye temperature and Grüneisen parameters are important to describe the thermal properties of a solid as the expansion coefficient. The temperature dependence of the Debye temperature is displayed in Figure 3a. One may observe that, within our treatment, $\theta_{D}$ significantly decreasing with T, where it goes down from 421 K to 360 K when the temperature rises from 0 K to 700 K. It is a known fact that the Debye temperature is an indicator of chemical bond strength and material hardness. The higher this quantity is, the stronger the chemical bonds and the larger the hardness will be. Thus, our results suitably point at the weakening of atomic cohesion with temperature [31]. Moreover, the higher the Debye temperature of the material is, the greater its thermal conductivity turns out to be, as shown in Ref. [32], which is something that we shall address below in more detail.

The Grüneisen parameter (γ) is used to describe the anharmonic oscillation of atoms in the structure about the equilibrium position (γ = 0 for harmonic oscillation) [33,34]. The evolution of γ with temperature is illustrated in Figure 3b, with values that rank from 2.34 to 2.54. Accordingly, this parameter increases with temperature in a linear-like variation, with a very modest slope in the order of 10${}^{-4}$.

As rightfully stated by Wehrl, entropy is one of the most important quantities in physics. This crucial concept of thermodynamics and statistical mechanics relates to the macroscopic and microscopic aspects of nature and determines the behavior of macroscopic systems [35]. Figure 4a shows the entropy of the system under concern, calculated as a function of temperature. It can be seen that the entropy of CZGS rapidly increases with temperature, with its value at room temperature equal to 212.17 J·mol${}^{-1}$·K${}^{-1}$. In close relation, Figure 4b contains the variation of the specific heat capacity $C_{v}$ of CZGS, which can provide an insight into the vibrational properties of the material [36]. We can observe a sharp increase in $C_{v}$ with a temperature up to 300 K. However, at high temperatures, the specific heat capacity reaches the well-known value named the Dulong–Petit limit [37]. In fact, the calculated Cv at room temperature is 181.9 J·mol${}^{-1}$·K${}^{-1}$ and the Dulong–Petit limit for CZGS is 198.28 J·mol${}^{-1}$·K${}^{-1}$.

### 2.3. Carrier Mobility and Relaxation Time

With regard to transport properties, our study of carrier mobility in kesterite CZGS was carried out within the deformation potential theory, combined with the effective mass approximation proposed by Bardeen and Shockley [38]. In this context, the carrier’s mobility, μ, is expressed as follows:(1)$$\mu =\frac{\sqrt{8\pi }\cdot e.\hbar^{4}\cdot C}{3.E_{d}^{2}\cdot {\left(k_{B}\cdot T\right)}^{3/2}\cdot {\left(m^{*}\right)}^{5/2}}$$
where *e* is the elementary charge, *C* is the elastic constant along the strain direction, $E_{d}$ is the deformation potential, $m^{*}$ is the effective mass of the carrier, and $k_{B}$ is the Boltzmann constant. The deformation potential can be calculated as $E_{d}=\frac{\partial E_{edge}}{\partial \delta }$, where $\partial E_{edge}$ indicates the change in the conduction band minimum (valence band maximum) for the electron (hole), induced by strain ∂δ. Furthermore, the strain δ is given by δ = $\frac{a}{a_{0}}-1$, where $a_{0}$ is the equilibrium lattice parameter, and a its corresponding value under deformation. The elastic constant C can be obtained using
$$C=\frac{1}{V_{0}}\frac{\partial^{2}E_{s}}{\partial \delta },$$
where $V_{0}$ is the volume of the structure at the equilibrium state and $E_{s}$ stands for the total energy of the system.

After acquiring the carrier’s mobility, it is straightforward to obtain the relaxation time of the carrier using $\tau =\mu |m^{*}|e^{-1}$. The effective masses values used in our calculation are $m_{h}^{*}=0.91m_{0}$ and $m_{e}^{*}=0.22m_{0}$ which were reported by Liu et al. [39]. The calculated mobilities and relaxation times for electrons and holes in the compound of interest are represented in Figure 5 as functions of temperature. It is readily apparent that the electron and hole mobility of CZGS decrease with $T^{-3/2}$ dependence, and the same trend is observed for the relaxation time. The value of the electron and hole mobilities at room temperature are 17,380.3 cm${}^{2}$V${}^{-1}$s${}^{-1}$ and 90.2 cm${}^{2}$V${}^{-1}$s${}^{-1}$, respectively. Additionally, the relaxation times of CZGS are in the order of 10${}^{-14}$ in all temperature ranges. Large departures between the hole and electron mobilities and relaxation times have to do—at least partially—with the big difference between their effective masses, where the isotropic hole effective mass is ∼0.91$m_{0}$ and the electron effective mass is 4 times lower with ∼0.22$m_{0}$. It is worth mentioning that the large room temperature electron mobility obtained for CZGS clearly surpasses the values reported for most IV and III–V semiconductors (with the exception of In-based ones) and is also greater than those recently calculated for some relevant II–VI compounds [40]. However, it must be borne in mind that our evaluation does not include other mobility-limiting scattering mechanisms such as electron–optical phonon interactions, which are particular relevant at higher temperatures.

For a complete description of the charge carrier properties, we also calculated the exciton binding energy with the Wannier–Mott formula [41]:(2)$$E_{b}=\frac{R_{y}\cdot m_{e}\cdot m_{h}}{m_{0}\left(m_{e}+m_{h}\right)\varepsilon_{0}^{2}},$$
where $R_{y}$ = 13.6057 eV is the Rydberg energy constant, $m_{e}$ and $m_{h}$ are the isotropically averaged electron, and hole effective masses are the electron free mass and $\varepsilon_{0}$ is the static dielectric constant. As shown in Table 1, the calculated exciton binding energy and static dielectric constant are 59.03 meV and 5.88 meV, respectively. The latter has been evaluated from the interband-related optical response investigated in a previous work [17]. There, a value of 2.42 was found for the CZGS static index of refraction. Such a result compares quite favorably with the experimentally reported one of 2.65, obtained at a lowest incident photon energy of 1.4 eV through variable angle spectroscopic ellipsometry [42].

### 2.4. Thermoelectric Properties

As already mentioned, the study of thermoelectric properties of kesterite Cu${}_{2}$ZnGeS${}_{4}$ is one of our main concerns here. To investigate the performance of CZGS as a thermoelectric material, it is necessary to evaluate the dimensionless factor called the figure of merit, ZT, which can be calculated from the Seebeck coefficient (*S*), the electrical conductivity (σ), the total thermal conductivity ($\kappa =\kappa_{e}+\kappa_{l}$) and the temperature, *T*, through the following expression
$$ZT=\frac{S^{2}\sigma T}{\kappa }$$

All the transport coefficients (*S*, σ and $\kappa_{e}$ ) are determined by solving the Boltzmann transport equation. However, the lattice thermal conductivity ($\kappa_{l}$) of CZGS is needed. In this work, the value of this quantity is predicted using the model by Slack model [43,44] where $\kappa_{l}$ is expressed as:(3)$$\kappa_{l}=\frac{2.4310^{-8}M_{a}\cdot \theta_{D}^{3}\delta }{\left(\gamma^{2}-0.514\gamma +0.228\right)\cdot T\cdot n^{2/3}}$$

In this expression, γ and $\theta_{D}$ are the Grüneisen parameter and Debye temperature, respectively, $M_{a}$ is the average atomic mass, $\delta^{3}$ is the volume per atom, *T* is the temperature and *n* is the number of atoms in the primitive cell. The calculated *S*, σ, $\kappa_{e}$, and $\kappa_{l}$ as functions of temperature are presented in Figure 6.

Desirable thermoelectric materials should present a high Seebeck coefficient, high electrical conductivity, and low thermal conductivity in order to directly and efficiently convert unused heat into electricity [23]. It is clear from Figure 6a that the Seebeck coefficient of CZGS is higher than 140 µV·K${}^{-1}$ in the entire temperature range from 300 K to 700 K. We note that the value of the Seebeck coefficient at room temperature is, in our case, 149.4 µV·K${}^{-1}$, and the maximum value is observed at 700 K to be 192.6 µV·K${}^{-1}$. Moreover, the observed maximum value is close to 230 µV·K${}^{-1}$), which has been confirmed experimentally and theoretically that many good thermoelectric materials with high thermoelectric conversion [45,46].

The electrical conductivity of kesterite CZGS is illustrated in Figure 5b. CZGS exhibits an electrical conductivity with a magnitude of $10^{5}$
$\Omega^{-1}$m${}^{-1}$ in the whole temperature range. It has been reported as a good candidate for thermoelectric materials. From Figure 6b, we can observe that the electric conductivity decreases from 4.1 ×$10^{5}$
$\Omega^{-1}$m${}^{-1}$ at 300 K to 2.2 ×$10^{5}$
$\Omega^{-1}$m${}^{-1}$ at 700 K.

According to Figure 6c, which presents the computed and the experimentally measured lattice thermal conductivity, it is possible to notice an enormous underestimation of the lattice thermal conductivity when it is calculated with the Slack model. Therefore, we will consider the experimental values for $\kappa_{l}$ in a further discussion. The comparison between $\kappa_{l}$ and the electronic thermal conductivity is given in Figure 6d. It allows noticing that, for the temperature range below 400 K, the computed electronic thermal conductivity of CZGS is lower than the lattice contribution to thermal conductivity. This means that, for a temperature higher than 400 K, the dominant thermal conducting mechanism is the migration of free electrons. The electrical part of the thermal conductivity increases with the temperature, whilst the inverse trend is observed for the lattice thermal conductivity. One may notice that our results reveal good features for thermoelectric conversion, according to the aforementioned criteria.

Another important quantity for a thermoelectric material is the so-called power factor. It can be calculated in the form $PF=S^{2}\sigma$. A large PF means that the thermoelectric device could deliver large voltage and current as output. The calculated PF of CZGS is presented in Figure 7a. CZGS at room temperature has a PF of 9.4 mW·m${}^{-1}$·K${}^{-1}$, which is higher, for instance, than SnS${}_{x}$Se${}_{x-1}$, known as low cat material with a high thermoelectric performance [47]. The PF of CZGS is decreasing with temperature to 8.26 mW·m${}^{-1}$·K${}^{-1}$ at 700 K.

The thermoelectric performance of CZGS can be measured after acquiring all the necessary parameters. Excellent material for converting the heat to electricity should have at least a ZT≥ 1. The calculated figure of merit dependence with temperature is illustrated in Figure 7b. As can be seen, ZT increases with the rising temperature. CZGS presents a moderate value of ZT at room temperature, which is 0.36 (up to 1.36 at 700 K). Such high ZT values observed for CZGS can be explained by the ultra-low lattice thermal conductivity estimated by the Slack model. In this sense, it is worth commenting that both the PF and ZT obtained for kesterite CZGS for a temperature higher than 550 K are close to those reported for a wide variety of thermoelectric materials. In fact, updated experimental and theoretical achievements for seven kinds of materials, including BiTe series, SnSe series, CuSe series, multicomponent oxides, half-Heusler alloys, organic–inorganic composites, and GeTe/PbTe series have been reviewed with regard to their thermoelectric behavior [47]. In practically all cases, kesterite CZGS compares favorably in what concerns PF and ZT values. This confirms the potential of CZGS as a promising material for thermoelectric applications.

## 3. Materials and Methods

In this work, we investigated the structural, electronic, thermodynamic, and thermoelectric properties of Cu${}_{2}$ZnGeS${}_{4}$ from the first principles, using the full-potential linearized augmented plan-wave (FPLAPW) [48] method implemented in the Wien2k code [49], to solve the Kohn–Sham equation within the framework of density functional theory (DFT). For the exchange and correlation potential, Tran-Blaha modified Becke and Johnson’s potential combined with the Hubbard potential U (mBJ + U). Even though mBJ is a GGA bandgap correction, it is ineffective in obtaining a bandgap that is near to the experimental data in our case. As a result, we correlate it with the Hubbard potential [50,51]. We chose this DFT approach because the usual DFT methods (LDA and GGA) are known for underestimating the bandgap energy [52]. DFT + U (mBJ + U) is a method for correcting the bandgap underestimate of conventional DFT that is commonly utilized in theoretical research [53]. The Hubbard potential U is a semi-empirical parameter that may be used in DFT calculations for transition metal elements with a high correlation [51]. We employed Hubbard’s potential to correctly modify the electrical behavior of Cu ([Ar] 4s1 3d10), Zn ([Ar] 4s2 3d10), and Ge ([Ar] 4s2 3d10 4p2) d-orbital atoms.

After the convergence tests, the value of the product of R${}_{mt}$ and K${}_{max}$ is chosen as 9, and 1000 k-points in the first Brillouin zone (BZ) are taken for all the calculations. Here, R${}_{mt}$ labels the smallest atomic sphere radius in the unit cell and K${}_{max}$ is the maximum modulus for the reciprocal lattice vector. The cut-off energy is set to −6 Ry, and the convergence of the self-consistent calculation is achieved when the energy difference between succeeding iterations is less than 10${}^{5}$. After examining the U values between 0.3 Ry and 0.52 Ry, the final U value used in our calculations was set to 0.48 Ry.

For thermodynamic calculations, the quasi-harmonic Debye model was used, and was implemented in the Gibbs2 package [54], within which the Debye temperature ($\theta_{D}$), Grüneisen parameter (γ), thermal expansion coefficient (α), entropy (S${}_{e}$), and constant-volume heat capacity (Cv) were determined through the following expressions:(4)$$\theta_{D}=\frac{\hbar }{k_{B}}{\left(\frac{6\cdot \pi^{2}n}{V}\right)}^{\frac{1}{3}}v_{0}$$
(5)$$\gamma =\frac{-\partial \ln\left(\theta_{D}\right)}{\partial \ln\left(V\right)}$$
(6)$$\alpha =\frac{1}{V}{\left(\frac{\partial V}{\partial T}\right)}_{p}$$
(7)$$S_{e}=-3\cdot nk_{B}\ln\left(1-e^{-\frac{\theta_{D}}{T}}\right)+4\cdot nk_{B}D\left(\frac{\theta_{D}}{T}\right)$$
(8)$$C_{v}=12\cdot nk_{B}D\left(\frac{\theta_{D}}{T}\right)-\frac{9\cdot nk_{B}\theta_{D}}{T\left(e^{\left(\frac{\theta_{D}}{T}\right)}-1\right)}$$

It is worth recalling that, within this approach, $\theta_{D}$ depends on the volume and consequently on temperature, also through the effective sound velocity $v_{0}$. In addition, in these expressions, D(x) is the Debye function, *T* is the absolute temperature, $k_{B}$ is the Boltzmann constant, and *n* represents the atomic volume density [29,55,56].

Transport calculations were performed by solving the Boltzmann transport equation within the rigid band and constant relaxation-time approximations, as implemented in BolzTrap2 [57].

## 4. Conclusions

We performed a systematic study of electric, thermodynamic, transport, and thermoelectric properties of kesterite Cu${}_{2}$ZnGeS${}_{4}$ through the first-principles density functional theory with Becke–Johnson potential, combined with the Hubbard potential U to treat the exchange and correlation effects. We first determined a band gap energy of 2.05 eV, which is consistent with experimental reports, and then we looked into its thermodynamic properties, obtaining values of 212.17 J·mol${}^{-1}$·K${}^{-1}$ for the entropy and 181.9 J·mol${}^{-1}$·K${}^{-1}$ for the constant-volume heat capacity. We determined that this compound exhibits rather large electron mobility but a much smaller hole one. The calculated electric conductivity is high, and the thermal one is significantly low compared with experimental reports. This allows us to conclude that the Slack model fails to provide the correct values in this case. On the other hand, the Seebeck coefficient in the room has a maximum value of 192.6 µV·K${}^{-1}$ at 700 K, which approaches the range desired for optimal thermoelectric conversion. Furthermore, Cu${}_{2}$ZnGeS${}_{4}$ at room temperature shows a power factor of 9.4 mW K${}^{-1}$ m${}^{-1}$, and a thermoelectric figure of merit of 0.36. These features indicate that this material could be applied in thermoelectric conversion.

## Figures and Tables

**Figure 1 ijms-23-12785-f001:**
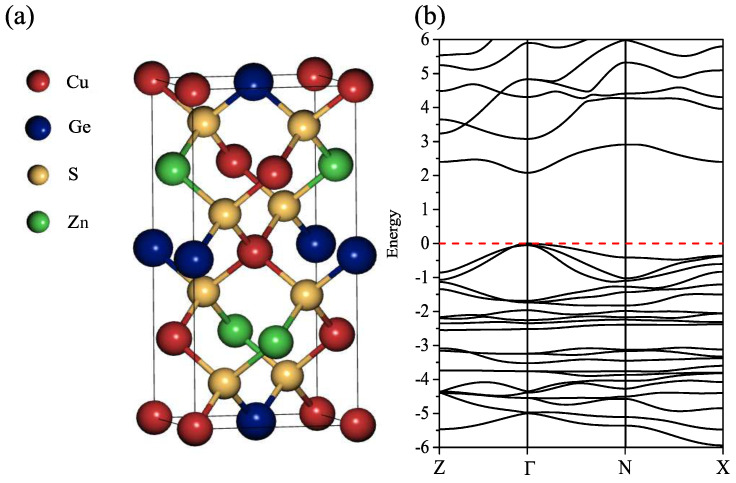
(**a**) The kesterite crystal structure; and (**b**) the calculated DFT-mBJ + U electronic band structure of kesterite Cu${}_{2}$ZnGeS${}_{4}$.

**Figure 2 ijms-23-12785-f002:**
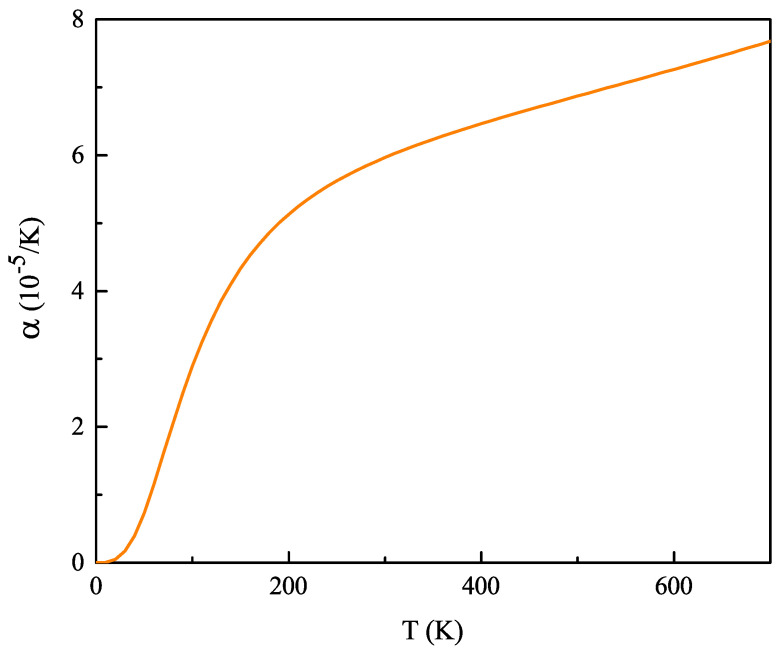
Thermal expansion coefficient as a function of temperature for bulk kesterite Cu${}_{2}$ZnGeS${}_{4}$.

**Figure 3 ijms-23-12785-f003:**
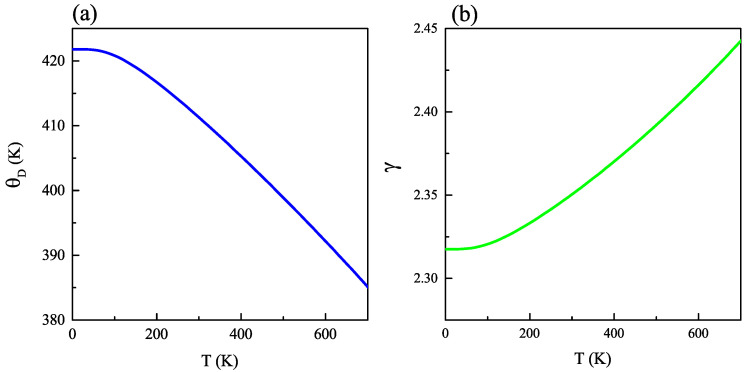
(**a**) Debye temperature and (**b**) Grüneisen parameter as functions of temperature in bulk kesterite Cu${}_{2}$ZnGeS${}_{4}$.

**Figure 4 ijms-23-12785-f004:**
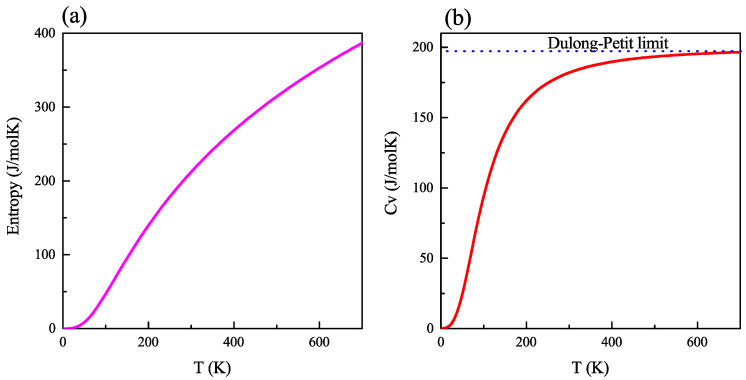
(**a**) Entropy and (**b**) constant-volume heat capacity as a function of temperature in bulk kesterite Cu${}_{2}$ZnGeS${}_{4}$.

**Figure 5 ijms-23-12785-f005:**
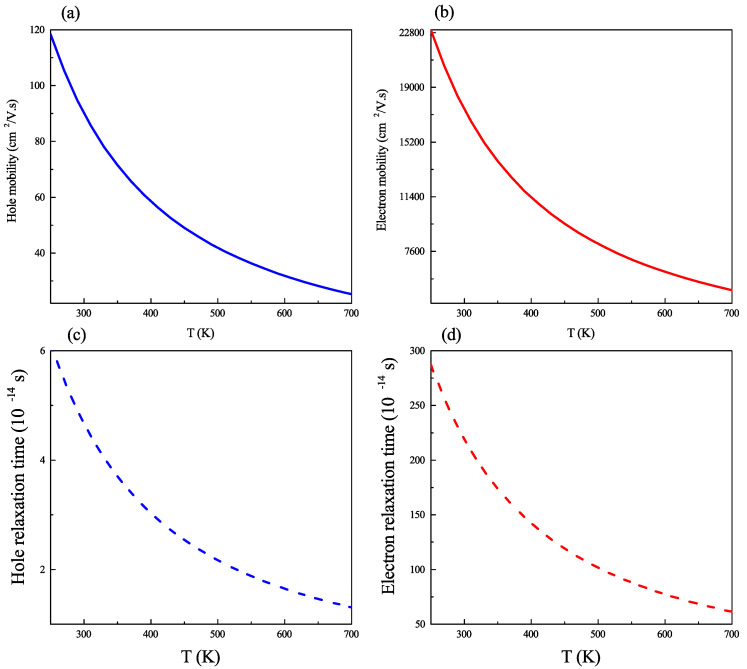
(**a**) Hole mobility; (**b**) electron mobility; (**c**) relaxation time of hole; and (**d**) relaxation time of electron as a function of temperature in kesterite Cu${}_{2}$ZnGeS${}_{4}$.

**Figure 6 ijms-23-12785-f006:**
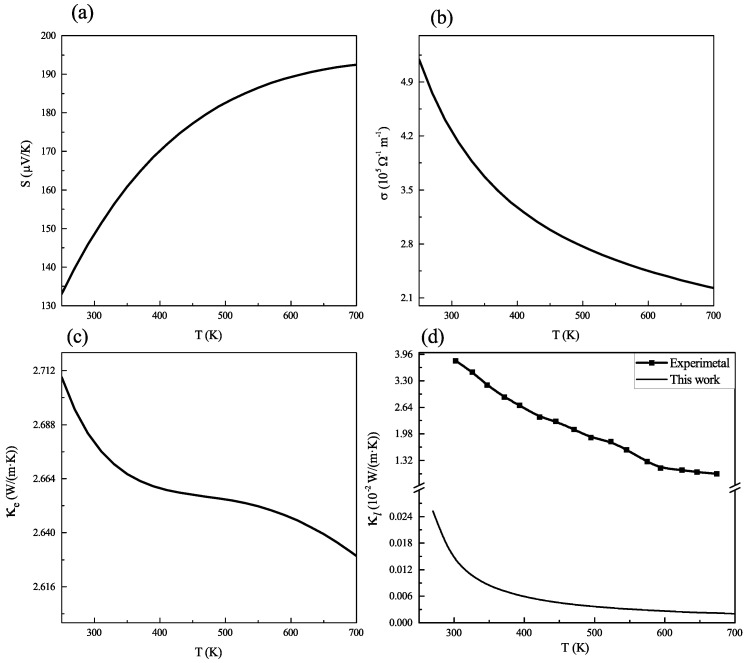
(**a**) Seebeck coefficient; (**b**) electrical conductivity; (**c**) electrical part of thermal conductivity; and (**d**) lattice thermal conductivity as functions of temperature kesterite Cu${}_{2}$ZnGeS${}_{4}$.

**Figure 7 ijms-23-12785-f007:**
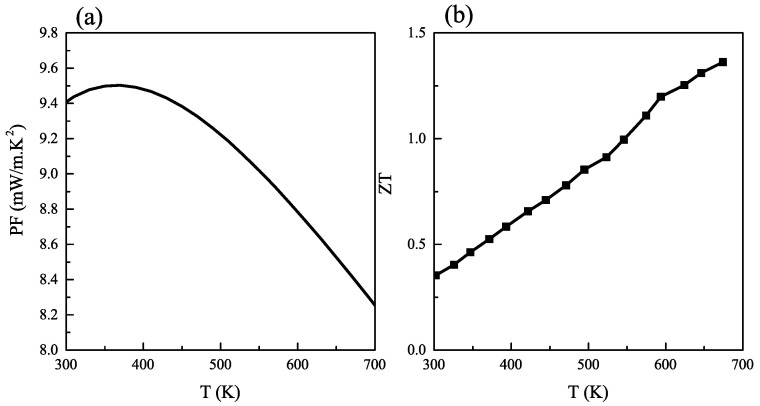
Variation of (**a**) power factor; and (**b**) figure of merit with temperature in kesterite Cu${}_{2}$ZnGeS${}_{4}$.

**Table 1 ijms-23-12785-t001:** The computed lattice parameters (a, c), band gap energy ($E_{g}$), Debye temperature ($\theta_{D}$), Grüneisen parameter (γ), thermal expansion coefficient (α), entropy ($S_{e}$), constant-volume heat capacity ($C_{v}$), electron mobility ($\mu_{e}$), hole mobility ($\mu_{h}$), electron relaxation time ($\tau_{e}$), hole relaxation time ($\tau_{h}$), Seebeck coefficient (S), electrical conductivity (σ), electron thermal conductivity ($\kappa_{e}$), power factor (PF), figure of merit (ZT) and exciton binding energy ($E_{b}$) at ambient temperature of CZGS. Whenever applied, the values are reported at room temperature.

Parameters	Values	Parameters	Parameters
a	5.281 (Å)	$\mu_{e}$	17,380.3 (cm${}^{2}$V${}^{-1}$s${}^{-1}$)
c	10.51 (Å)	$\mu_{h}$	90.2 (cm${}^{2}$V${}^{-1}$s${}^{-1}$)
$E_{g}$	2.05 (eV)	$\tau_{e}$	218.8 × 10${}^{-14}$ (s)
α	5.9 × 10${}^{-5}$ (K${}^{-1})$	$\tau_{h}$	4.53 × 10${}^{-14}$ (s)
$\theta_{D}$	411 (K)	*S*	149.4 (µV·K${}^{-1}$)
γ	2.35	σ	4.1 × 10${}^{5}$ ($\Omega^{-1}$m${}^{-1}$)
$C_{v}$	181.9 (J mol${}^{-1}$K${}^{-1}$)	$\kappa_{e}$	2.68 (WK${}^{-1}$m${}^{-1}$)
$S_{e}$	212.17 (J mol${}^{-1}$K${}^{-1}$)	PF	9.4 (mW K${}^{-1}$ m${}^{-1}$)
$E_{b}$	59.03 (meV)	ZT	0.36

## Data Availability

Data are available upon request from the corresponding authors.

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
