# Peer review of "Ab Initio Study of Carrier Mobility, Thermodynamic and Thermoelectric Properties of Kesterite Cu2ZnGeS4"

_ijms, 2022, doi:10.3390/ijms232112785_

Round 1
Reviewer 1 Report
Review (recommended major revision)
In article, authors studied the Cu2ZnGeS4 material for its targeted applications in thermo-electronic end uses. Manuscript contains the extensive density functional theory (DFT) model calculations with some applied corrections, such as Hubbard correlation, present to account for band gap underestimation. Revise.
***attached review file***

Author Response
Dear Reviewer,
Thank you very much for your report. Please find attached our reply of your remarks.
Sincerely yours,
David Laroze

Reviewer 2 Report
The manuscript describes and discusses logically designed experiments and presents results that are expected to be of large interest for the scientific community. It is an interesting study with an interesting approach. The paper in the whole is well designed and results sound. Nevertheless, the manuscript needs a minor revision:
Point 1: In the introduction part should be more highlighted the main aim of the paper, and additionally, what is the novelty of carried research work.
Point 2: How do the Authors select the analytes? The rational of the choice of the selected biologically active compounds studied is missing and should be clearly discussed.
Point 3: Quality of the figures must be improved.
Author Response

(The authors gave the same response as above.)

Reviewer 3 Report
Article entitled Ab-initio study of carrier mobility, thermodynamic and thermoelectric properties of kesterite Cu2ZnGeS4 deals with the calculation of physical properties of inorganic substance usefull in energy conversion devices.
The article is suitable for Special Issue Nano-Materials and Methods 3.0. But I have some issues which must be fixed before further publication in IJMS.
Firstly, authors should thoroughly revise the English in the text.
Next, there is a problem with introduction section. The Introduction does not give the reader sufficient insight into the topic. It also lacks the motivation why it is good to make such calculations and determine physical parameters using model studies. Finally, there is no comparison with parameters obtained using physical measurements. And if they are not, authors can emphasize that motivation.
Next thing is the missing explanation of dual type of charge carrier transport. The author should explain this mechanism.
Which value of m0 was used for calculation of effective mass in the part with relaxation time? In the text this is not explained.
In Figure 6, Graph b – I have doubts that the values of conductivity are correct. This is also related to claims in conclusion. There is missing critical view. Because the conductivity of such kind material is usually in three or more orders lower.
The authors must soften the claims in the conclusions, because if they do not have a comparison with real measurements, the theoretical calculations remain only a hypothetical approximation with a certain degree of probability.
Author Response

(The authors gave the same response as above.)

Round 2
Reviewer 1 Report
Review (accept)
Comments have been mostly addressed, improving this interesting manuscript.
Reviewer 3 Report
The authors tried to improve the manuscript based on my comments. The manuscript is suitable for publication.